# Anti-Infective Treatment and Resistance Is Rarely Problematic with Eye Infections

**DOI:** 10.3390/antibiotics11020204

**Published:** 2022-02-06

**Authors:** Regis P. Kowalski, Shannon V. Nayyar, Eric G. Romanowski, Vishal Jhanji

**Affiliations:** 1Department of Ophthalmology, The Eye and Ear Institute, School of Medicine, University of Pittsburgh, 203 Lothrop Street, Pittsburgh, PA 15213, USA; romanowskieg@upmc.edu (E.G.R.); jhanjiv@upmc.edu (V.J.); 2The Charles T. Campbell Eye Microbiology Laboratory, Department of Ophthalmology, University of Pittsburgh Medical Center, The Eye and Ear Institute, University of Pittsburgh School of Medicine, 203 Lothrop Street, Room 642, Pittsburgh, PA 15213, USA; nayyars@upmc.edu

**Keywords:** ophthalmic anti-infectives, anti-infective susceptibilities, endophthalmitis, keratitis, conjunctivitis, blepharitis, ophthalmic diagnostic testing, eye infections

## Abstract

The treatment of eye infections is very different than treating other body infections that require systemic anti-infectives. Endophthalmitis, keratitis, conjunctivitis, and other ocular infections are treated with direct injection and with topical drops directly to the infection site. There are no anti-infective susceptibility standards to interpret treatment success, but the systemic standards can be used to guide ocular therapy if the concentration of anti-infective in the ocular tissue is assumed to be higher than the concentration in the blood serum. This *Perspective* describes: (1) eye infections, (2) diagnostics of eye infections, (3) anti-infective treatment of eye infections, (4) anti-infective resistance of ocular pathogens, and (5) alternative anti-infective delivery and therapy. The data, based on years of clinical and laboratory research, support the premise that ocular infections are less problematic if etiologic agents are laboratory-diagnosed and if prompt, potent, anti-infective therapy is applied. Anti-infective susceptibility should be monitored to assure continued therapeutic success and the possibility of new-found resistance. New delivery systems and therapies may be helpful to better treat future ocular infections.

## 1. Introduction

The treatment of eye infections is completely different than other parts of the body. Eye infections in general are not treated systemically but are treated with anti-infectives using topical drops and direct injection, which provide very high effective levels of anti-infectives into the ocular tissue [1]. (Note: Antibiotics and anti-infectives will be referred to as anti-infectives in this *Prospective*. Anti-infectives are man-made like the fluoroquinolones. Penicillin and cephalosporins are biologically produced antibiotics.) Whereas the CLSI susceptibility standards (Clinical and Laboratory Standard Institute) were established to interpret anti-infective susceptibility via the systemic route, there are no susceptibility standards to determine an in vitro to in vivo correlation between laboratory testing and patient recovery for eye infection treatment [2]. Previous studies have indicated that ciprofloxacin, ofloxacin, moxifloxacin, and besifloxacin may be effective against bacterial keratitis isolates on the basis of concentrations achieved in human corneal tissue [3,4,5]. Many clinical microbiology laboratories (if not all) will not report the anti-infective susceptibilities of ocular bacterial isolates using the systemic serum standards because of government (Federal, State) and organizational (CLIA (Clinical Laboratory Improvement Amendments), JCAHO (Joint Commission on Accreditation of Healthcare Organization), and CAP (College of American Pathology)) regulations. Susceptibility interpretation of eye bacterial isolates would be off-label and a contravention of laboratory standards and rules. Although anti-infective resistance has become a global setback with systemic infections, anti-infective resistance with ocular anti-infectives is less problematic.

Intuitive reasoning based on years of successful practice indicates that ocular treatment using topical anti-infectives, subconjunctival injections, and intravitreal injections have clear-cut advantages over systemic treatment [6]. Systemic treatment by oral and intravenous routes involve absorption of anti-infectives into the blood system to arrive at the site of infection at therapeutic safe serum concentrations. However, there is no delivery required via the vascular system for the direct treatment of ocular infections. High concentrations of anti-infectives above the serum standards are directly achieved at the ocular infection sites with topical and direct injection application. Although there are no ocular standards for interpreting susceptibility on the basis of topical and intravitreal treatment, the serum standards can be used to guide treatment on the basis of the reasonably assumed high concentration of anti-infectives in the ocular tissue. Systemic clinical microbiology laboratories could use the CLSI standards to guide effective ocular antibiotic treatment with the recognition of an important assumption that “Anti-infective concentrations achieved in the ocular tissues by the topical or direct injection are higher than the anti-infective concentrations achieved in the ocular tissues through oral or intravenous administration”.

This *Perspective* will describe (1) eye infections, (2) diagnostics of eye infections, (3) anti-infective treatment of ocular infections, (4) anti-infective resistance of ocular pathogens, and (5) alternative anti-infective delivery and therapy. The *Perspective* is based on the experiences of a certified independent clinical ophthalmic microbiology laboratory. (http://eyemicrobiology.upmc.com) (accessed on 25 January 2022).

## 2. Eye Infections

Eye infections predominately involve the aqueous, vitreous, cornea, conjunctiva, and eyelids. Other areas can be infected such as the lacrimal sac and the canaliculi. The location of the infected area (e.g., aqueous, vitreous, cornea, conjunctiva, eyelid) describes the diagnosis.

Bacteria that enter the eye and compromise the aqueous and/or vitreous cause inflammation and an intraocular infection known as endophthalmitis [7] The aqueous and vitreous have no colonizing bacteria. Any bacterial growth from an endophthalmitis culture is considered significant as a pathogen. These infections commonly occur after cataract surgery, other ocular surgeries, and ocular trauma. These infections can also originate systemically as an endogenous endophthalmitis. Figure 1 presents the distribution of bacterial endophthalmitis pathogens isolated from 2004–2018 at the Charles T. Campbell Eye Microbiology Laboratory at the UPMC Eye Center, University of Pittsburgh School of Medicine, Pittsburgh, PA, USA [8].

A bacterial infection of the avascular cornea (clear region at the front of the eye) is diagnosed as bacterial keratitis. The cornea, sclera, and conjunctiva have no colonizing bacteria. These regions may be contaminated with bacteria from the eyelid margin, but this contamination is soon eliminated by the host defense system of the conjunctival mucous membrane [9]. It must be noted that infectious keratitis differential can include fungi, viruses (herpes simplex virus (HSV), varicella zoster virus (VZV), and adenovirus), acanthamoeba, and microsporidia. Any presence of pathogenic bacteria cultured from the cornea should be suspected as the possible etiologic agent. Superinfection is not common but will be detected with pan testing. Figure 2 presents the distribution of infectious keratitis pathogens isolated from 2004 to 2018 at the Charles T. Campbell Eye Microbiology Laboratory at the UPMC Eye Center, University of Pittsburgh School of Medicine, Pittsburgh, PA, USA [8].

The cornea comprises several layers, with the epithelium as the top layer for protection. During corneal infection, the epithelium ulcerates, the cornea appears opaque, the eye displays redness and tearing with exudate, foreign body sensation may be present, there is pain, and the eye may become sensitive to light.

Infectious inflammation of the conjunctiva is generally due to bacteria and viruses. Bacterial conjunctivitis is generally self-limiting (1–3 days), but *Chlamydia trachomatis* and *Moraxella* infection may be extended if not properly treated [1,2,3,4,5,6,7,8,9,10]. Bacterial conjunctivitis is commonly due to *Streptococcus pneumoniae* and *Haemophilus influenzae* in children (pink eye), and *Staphylococcus aureus* in adults. Bacterial growth is abundant from the conjunctiva and eyelid margin. Adenovirus conjunctivitis is very contagious as live virus persists for 1–2 weeks. The extended immunologic symptoms involving the cornea may last for months to years. Adenovirus conjunctivitis is not blinding but symptoms due to immunogenic factors present with adenopathy, swelling, photophobia, excessive tearing, and foreign body sensation. Figure 3 presents the distribution of common infectious conjunctivitis pathogens isolated from 2004–2018 at the Charles T. Campbell Eye Microbiology Laboratory at the UPMC Eye Center, University of Pittsburgh School of Medicine, Pittsburgh, PA, USA [8].

## 3. Diagnostics of Eye Infections

In general, laboratory studies to diagnose ocular infection are based on the experience of the ophthalmologist. A first-year resident in ophthalmology is more likely to pan test for a wider range of pathogens, whereas a more experienced ophthalmologist may narrow the diagnosis to a few pathogenic groups. Many ocular infections may appear less benign in severity, with an early onset and a classic presentation. These infections are generally treated empirically without laboratory studies. Severe infections should undergo laboratory studies to confirm a diagnose and the assurance of proper therapy. (http://eyemicrobiology.upmc.com) (accessed on 25 January 2022)

Cultures to confirm endophthalmitis are of intraocular samples obtained from the aqueous and vitreous (or both) using a syringe and needle. The collected samples (a few drops) are routinely plated on trypticase soy agar supplemented with 5% sheep blood (5% SB), an aerobic chocolate agar, an anaerobic chocolate agar, a Sabouraud dextrose agar supplemented with gentamicin (SAB), and an enriched thioglycolate broth. A total of 5% SB and chocolate agar plates are incubated at 37 °C in 6 % CO_2_, and SAB at 30 °C. For a rapid detection, a few drops of vitreous are placed on glass slides for direct examination by Gram and Giemsa stain to observe for microorganisms and cytology. It is our experience that a vitreous culture is more diagnostic than an aqueous culture. In severe cases, extracted vitreous after a vitrectomy procedure is concentrated by centrifugation. The centrifuged pellet undergoes laboratory studies. Vitrectomy samples are very diagnostic, especially in cases of endogenous fungal and bacterial endophthalmitis. PCR is used to detect intraocular inflammation due to viruses (HSV, VZV, CMV (cytomegalovirus), EBV (Epstein Barr virus), etc.) and toxoplamosis. PCR using 16s rRNA and 18s rRNA has not been validated for ocular clinical samples but is frequently used to identify isolated bacteria and fungi from culture.

For keratitis, corneas are cultured directly using spatulas or jeweler’s forceps and planting the collected samples on 5% SB, an aerobic chocolate agar, a mannitol salt agar (selective for Staphylococci), and SAB. Collected samples are also placed on glass slides for direct examination by Gram and Giemsa stains to observe for microorganisms and cytology. An expanded differential may include acanthamoeba culture (non-nutrient agar overlaid with *Enterobacter aerogenes*), acanthamoeba PCR, and viral PCR for HSV and VZV.

Cultures of the conjunctiva and eyelid are collected with soft-tipped applicators and placed on the same culture media as used for keratitis cultures. PCR is used to detect adenovirus and HSV DNA, while nucleic acid amplification testing (NAAT) is used to detect *Chlamydia* ribosomal RNA [11,12,13,14]. Point of care of testing for adenovirus is not reliable and is not used by our laboratory for diagnosing adenovirus infection. It is misleading and often not definitive, providing false-negative results.

## 4. Anti-Infective Treatment of Ocular Infections

The first step for successful topical and intravitreal treatment of bacterial ocular infections is determining and confirming the best anti-infective to administer. Minimum inhibitory concentrations (MICs) determined with CLSI interpretation can guide (not confirm) the best in vitro anti-infective for combating bacterial disease in the eye.

All susceptibility testing is performed on pure, isolated colonies from 24 h culture growth on solid medium. Testing from primary liquid isolation medium can be done, but later confirmation from solid medium is necessary. Criteria for antibiotic testing is based on bacterial species, isolation site/diagnosis, and special requests. All bacterial species with known pathogenicity (*Pseudomonas aeruginosa, Staphylococcus aureus, Streptococcus pneumoniae,* etc.) are tested for anti-infective susceptibility. All bacteria species isolated from cornea and intraocular specimens, regardless of the species, should also be tested. Staphylococcal species isolated from the lids in chronic and severe blepharitis cases can be tested along with Staphylococcus species isolated from the lids in patients diagnosed with marginal ulcers, catarrhal ulcers, and phlyctenular disease.

MIC testing for ocular bacterial isolates is a specialty in which specific anti-infectives are chosen. Special batteries using E-tests (Liofilchem, Abruzzi, Italy) can be conveniently utilized. The following table comprises antibiotics that should be tested with different ocular pathologies.

### 4.1. In Vitro Antibiotic Susceptibility Testing Batteries

#### 4.1.1. Endophthalmitis

Gram-positive bacteria: vancomycin, cefoxitin (*Staphylococcus aureus* only).

Gram-negative bacteria: amikacin, ceftazidime.

#### 4.1.2. Keratitis

Gram-positive bacteria: vancomycin, moxifloxacin, gentamicin, cefazolin, cefoxitin (*Staphylococcus aureus* only).

Gram-negative bacteria: tobramycin, ciprofloxacin, polymyxin B, ceftazidime.

#### 4.1.3. Conjunctivitis

Gram-positive bacteria: moxifloxacin, cefoxitin (*Staphylococcus aureus* only).

Gram-negatives bacteria: tobramycin, ciprofloxacin, polymyxin B, moxifloxacin.

Patients presenting with endophthalmitis are treated empirically with vancomycin by injecting 1 mg of vancomycin in a 0.1 mL volume directly into the vitreous for Gram-positive infections. For Gram-negative bacterial infections, amikacin (0.4 mg/0.1 mL) or ceftazidime (2.00–2.25 mg/0.1 mL) is intravitreally injected. For empiric Gram-positive and Gram-negative bacterial coverage, all cases of endophthalmitis are injected with vancomycin and ceftazidime or amikacin. It must be noted that the human eye has an average vitreous volume of 4.0 mL. After intravitreal treatment, this would indicate that the anti-infective concentrations to be 250 µg/mL of vancomycin, 500 µg/mL of ceftazidime, or 100 µg/mL of amikacin. This is a large amount of anti-infective and is largely over the amount indicated by the serum standards to denote susceptibility [15,16,17]. Although amikacin and ceftazidime are not first-line anti-infectives for treating Gram-positive endophthalmitis, either may provide additional coverage at increased concentrations.

Bacterial keratitis is empirically treated topically with fortified cefazolin (50 mg/mL) for Gram-positive infections and fortified tobramycin (14 mg/mL) for Gram-negative bacterial infections. European alternatives would be topical fortified gentamicin (15 mg/mL) and fortified cefuroxime (50 mg/mL) [18]. Anti-infectives are referred to as fortified because they are not commercially available and must be prepared by a pharmacy. Prior to anti-infective topical treatment, corneas must be cultured for identifiable bacteria to confirm appropriate treatment. In order for the most appropriate treatment by the most potent anti-infective to be assured, treatment may be adjusted. Keratitis due to methicillin-resistant *Staphylococcus aureus* (MRSA) could be changed from cefazolin to topical fortified vancomycin (50, 25, 20 mg/mL). On the basis of keratitis severity and attendings experience, commercial formulations such as 0.5% moxifloxacin, 0.3% gentamicin, and bacitracin (500 units/gram) can also be used to treat MRSA [19,20,21]. For Gram-negative corneal infections, topical fortified tobramycin could be combined with 0.3% ciprofloxacin, or polymyxin B (10 units/mL). Ciprofloxacin has been reported to be a potent anti-infective for *Pseudomonas aeruginosa* infection [22,23,24]. Although not considered a Gram-negative anti-infective, moxifloxacin is often prescribed to treat infiltrates caused by contact lens wear. Besifloxacin is a commercial third-generation fluoroquinolone anti-infective formulated as an ophthalmic suspension in Durasite^®^ to treat bacterial keratitis [25]. Wu reported that Durasite^®^ was not bioactive to bacteria but it broke up biofilm, which may allow better contact between bacteria and anti-infective. The dispersed biofilm matrix may also contain immunologic matter [26]. Topical amikacin at 14 mg/mL may be advantageous for treating mycobacteria keratitis.

Conjunctivitis can be treated topically with commercial formulations to speed up recovery. Generic formulations tend to be less expensive to treat a disease that may spread to others in close quarters such as a school or daycare center. Sulfacetamide (10%), and polymyxin B (10,000 units)/trimethoprim (1 mg) per mL combination are commonly used. Erythromycin ointment 0.5% can be used to treat conjunctivitis and blepharitis. Although ointment application can be messy, dosing can be decreased especially for night-time treatment.

As indicated, susceptibility testing of ocular bacterial isolates for topical and intravitreal treatment can be guided by the CSLI serum standards, but many important ocular anti-infectives have no systemic susceptibility interpretations for guidance. Cefazolin, a first-line cephalosporin for keratitis, bacitracin (frequently used to treat *Staphylococcus aureus* infections, including MRSA), erythromycin, neomycin, sulfacetamide, ofloxacin, levofloxacin, and besifloxacin have no standards for the lists of pathogens that infect the eye. Many anti-infectives do not reach high concentration levels in the serum, but high levels may be achieved in the ocular tissues with topical treatment. Penicillin is not used for ocular treatment because the resistance interpretation of the standard is so low due to the interpretation for systemic treatment [27,28].

Table 1 provides the descriptive statistics of anti-infectives used to treat common ocular infections. These are the only anti-infectives used for ocular infections that have standards by CLSI. Without interpretations, only educated observations can be surmised. For keratitis, in vitro anti-infective susceptibility testing indicates that topical vancomycin would cover *Staphylococcus aureus* (including MRSA), but the use of other anti-infectives must be confirmed with testing. Although ceftazidime is not commonly used for topical *Pseudomonas aeruginosa* and Gram-negative bacterial coverage, in vitro testing indicates effective treatment. Coagulase negative Staphylococcus is a questionable keratitis pathogen with benign clinical features in which vancomycin appears to have the best coverage [29]. *Streptococcus viridans* group is best topically covered by moxifloxacin. It appears that Gram-positive bacteria (including *Streptococcus pneumoniae*, *Streptococcus pyogenes*, *Enterococcus species*, *Bacillus species*, diphtheroids, etc.), as expected, are covered by vancomycin.

Infectious conjunctivitis is usually treated with inexpensive anti-infectives due to its self-limiting nature. Treatment indication does not allow for interpretative standards, and treatment with fortified formulations is generally not necessary.

Over 90% of endophthalmitis is due to Gram-positive bacteria [7,15], which are susceptible to vancomycin. The half-life of vancomycin [16] in the vitreous is 25 h, and a concentration of 200–250 µg/mL would assure effective treatment.

## 5. Anti-Infective Resistance of Ocular Bacterial Isolates

True anti-infective resistance would involve acquiring resistance, spreading to others, and having no effective alternative anti-infective treatment [6]. This does not exist for treating ocular infections. The literature has many in vitro reports of ocular bacteria resistance to anti-infectives, but the resistance in every report is based on interpretation using systemic susceptibility standards [6,30]. The ARMOR (Antibiotic Resistance Monitoring in Ocular Microorganisms) and TRUST (Ocular Tracking Resistance in U.S. Today) studies were both based on serum standard interpretations and did not include cefazolin, sulfacetamide, or bacitracin, nor other anti-infectives used commonly for ocular infections [30]. Anti-infective resistance is not a problem when treating eye infections because anti-infective concentrations in the ocular tissue are extremely high due to topical dosing or direct injection. Bacterial keratitis and endophthalmitis are not spread from patient to patient. These infections are introduced independently from the host and not from other patients, although an ophthalmologist could be an infecting vector via surgery. Bacterial conjunctivitis can be spread from person to person, but the disease is most often self-limiting. There are reports and it is common knowledge that *Streptococcus pneumoniae* can be resistant to penicillin [31] and beta lactam anti-infectives, but this class of infectives is never used to treat bacterial conjunctivitis.

Fluoroquinolone (FQ) anti-infectives have become quite popular for ocular surgical prophylaxis, as well as for topical treatment for bacterial keratitis and conjunctivitis. Ciprofloxacin and ofloxacin are second-generation FQs, besifloxacin is a third-generation FQ, and moxifloxacin and gatifloxacin are fourth-generation FQs [29,31]. Against ocular isolates of endophthalmitis, keratitis, and conjunctivitis, moxifloxacin, on the basis of MICs, is the most potent FQ for Gram-positive bacteria, and ciprofloxacin is the most potent FQ for Gram-negative bacteria [22,23,24].

The key difference in the FQ generations is the resistance mechanisms. Bacterial resistance to the fluoroquinolones is due to mutations to the DNA gyrase and topoisomerases, as well as efflux pumps [32,33]. The second-generation fluoroquinolones require a single mutation for resistance, and the third and fourth generations require double mutations. There is inference of increasing FQ resistance, and there are reports of ciprofloxacin resistance in the ophthalmic literature according to systemic interpretation [34,35,36,37,38]. In vivo studies have demonstrated that resistance predicted by the in vitro systemic standards may not be accurate. Gatifloxacin and levofloxacin at commercial formulations successfully treated MRSA deemed resistant to gatifloxacin (Figure 4) and levofloxacin (ME) in rabbit keratitis models [37,39].

Figure 4 presents the mean Log_10_ corneal colony counts of rabbits intracorneally infected with MRSA deemed resistant to gatifloxacin (MIC = 12 µg/mL) that were topically treated with 0.3% gatifloxacin (Zymar^®^), 50 mg/mL fortified cefazolin, 50 mg/mL fortified vancomycin, and saline. The maroon line bisecting the graph represents the mean Log_10_ MRSA colony counts present in the corneas at the onset of therapy. Only 0.3 % gatifloxacin demonstrated a bactericidal decrease in MRSA colony counts compared to the onset of therapy control [37].

Theraja and Durrani reported in separate clinical studies that moxifloxacin was used successfully to treat MRSA eye infections that were considered to have moxifloxacin resistance [19,20]. These data were supported by Chang in the treatment of *Staphylococcus aureus* keratitis [21].

Treatment of MRSA ocular infection is not the challenge or threat that occurs in systemic treatment. Topical administration of fortified vancomycin (20–50 mg/mL) [40] provides high concentrations into the corneal tissue. Chang reported other anti-infectives (Figure 5) could potentially be used to treat MRSA keratitis [21]. Although cefazolin is contraindicated for treating MRSA infection, Chang’s study indicated possible successful outcomes. Administering high topical doses of 50 mg/mL of fortified cefazolin may still allow binding of the anti-infective to the penicillin binding proteins 2a along with the other pBps. Romanowski reported the efficacy of fortified cefazolin (Figure 4) using a rabbit keratitis model [37]. Romanowski reported that fortified penicillin at 6% topical dosing was also effective in a rabbit keratitis model to treat MRSA and penicillin-resistant *Staphylococcus aureus* [28]. Penicillin is not used in keratitis treatment because of the low MIC resistant interpretation for many isolates of *Staphylococcus aureus* and the possibility of allergic anaphylaxis. Topical allergic anaphylaxis is assumed but has not been reported in the literature for topical ocular penicillin administration. Figure 5 indicates that other topical ocular anti-infectives may be effective for treating MRSA. These anti-infectives were tested using the disk diffusion susceptibility method and interpretation using CLSI or manufacturers standards.

## 6. Alternative Anti-Infective Delivery and Therapy

The delivery of anti-infectives to infected sites (cornea, conjunctiva, and intraocular) have not drastically changed over the years. Standard therapy is either topically (administering anti-infectives directly to ocular surface in a drop) or directly injecting into the vitreous using a syringe and needle. These approaches have been effective but improvements to provide more localized anti-infectives at a reduced frequency have generated interest [41]. Viscous formulations in the forms of adding viscosity to aqueous solutions and suspensions have been developed for azithromycin [42], besifloxacin [25], and moxifloxacin (Moxeza^®^) [43] to improve contact time between the anti-infectives and ocular surface. Ointments containing anti-infectives have long been used to decrease dosing and increase contact time of anti-infective to the ocular surface. Contact lenses have been combined with anti-infectives to provide immediate or controlled release of anti-infectives to the ocular tissues [44,45,46,47].

There has been great interest for inserting materials into the cul-de-sac to gradually release anti-infectives over a longer period to treat eye infections. This would substitute for the use of frequent topical drops. Thus far, there has been no clinical studies, but Mammen introduced the potential use of thermo-responsive controlled-release microspheres loaded with moxifloxacin to prevent endophthalmitis in a rabbit model [48]. Karlund presented the use of collagen-binding domains to deliver molecules to the cornea. These domains would allow longer contact of anti-infectives to the ocular tissue, allowing more effective treatments of infection [49]. Hot-melt extruded inserts may also provide future once-a-day treatment for ocular infections [50]. Nanoparticles in an emulsion has been tested to treat bacterial keratitis with ciprofloxacin [51]. Shanks proposed the use of predatory bacteria to eliminate ocular pathogens as method to treat ocular infections [52,53]. Photoactivated chromophore for collagen cross-linking of the cornea [54] and rose Bengal photodynamic antimicrobial therapy [55] have been tested to treat severe, progressive infectious keratitis.

It must be noted that inserts and nanoparticles have not been clinically used to treat intraocular infections. There is great interest in using these technologies to treat non-infectious ocular disease such as glaucoma and age-related macular degeneration. Systemic treatment is not an optimal treatment for ocular infections. Thus, in contrast, systemic nanoparticles have been suggested to be a possible future treatment to treat ocular infections [56].

## 7. Summary

Although the treatment of ocular infections appears to be less problematic, anti-infectives need to be monitored for continued success. The causative etiologic agents of infections need to be identified by culture, and susceptibility testing needs to confirm treatment success. Empiric anti-infective treatment may need to be adjusted to assure potent therapy. Treating ocular infections with the most potent anti-infectives will prevent anti-infective resistance and provide a favorable prognosis for patient care.

Research may one day provide better delivery systems and alternative therapies for treating ocular infectives by providing larger anti-infective concentrations using less dosing for optimal patient compliance.

## Figures and Tables

**Figure 1 antibiotics-11-00204-f001:**
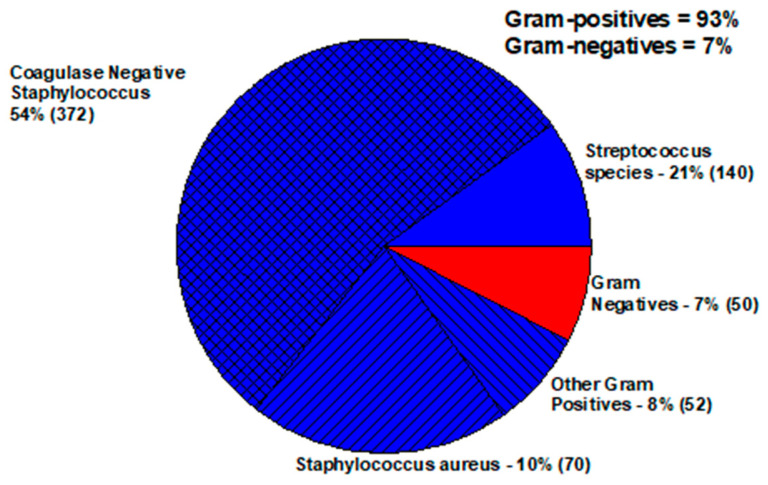
Bacterial pathogens of endophthalmitis.

**Figure 2 antibiotics-11-00204-f002:**
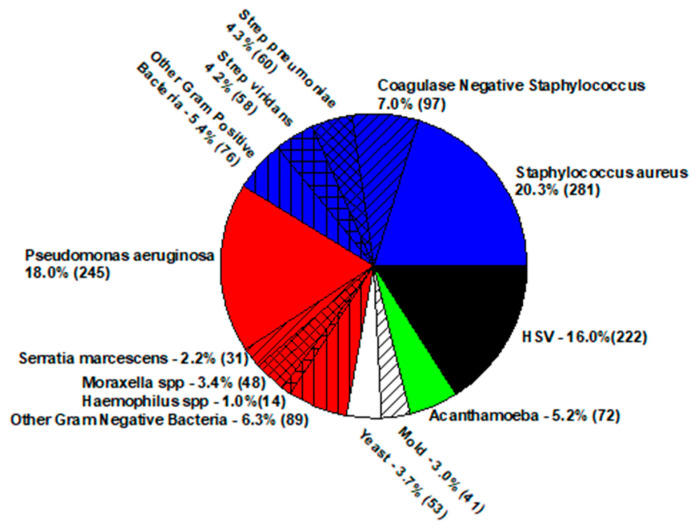
Distribution of infectious keratitis pathogens.

**Figure 3 antibiotics-11-00204-f003:**
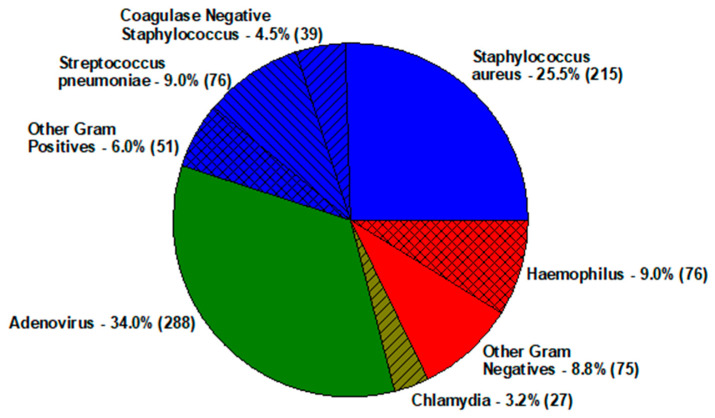
Distribution of infectious conjunctivitis pathogens.

**Figure 4 antibiotics-11-00204-f004:**
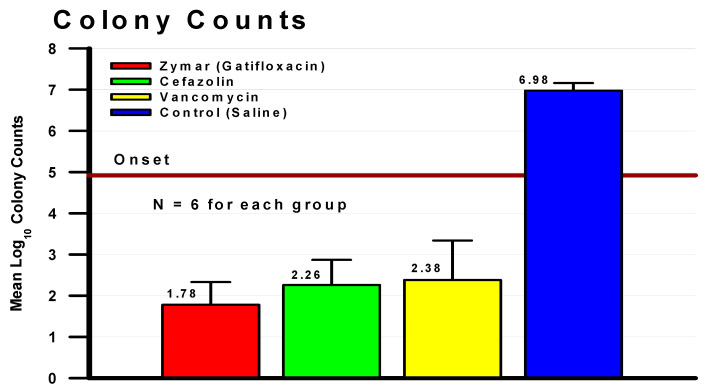
Topical treatment of MRSA keratitis with gatifloxacin, cefazolin, and vancomycin.

**Figure 5 antibiotics-11-00204-f005:**
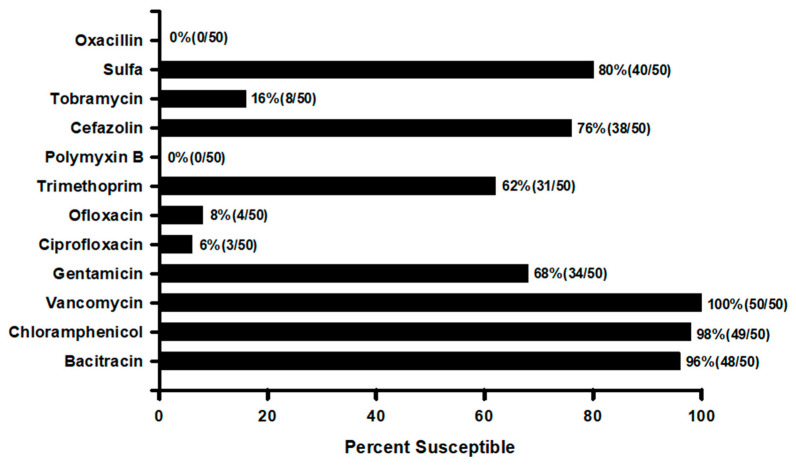
In vitro susceptibility of ocular MRSA to common topical anti-infectives.

**Table 1 antibiotics-11-00204-t001:** MICs (µg/mL) of bacteria isolated from eye infections to common anti-infectives (January 2020 to September 2021). List contains the serum standards that may guide the treatment of ocular infections. (http://eyemicrobiology.upmc.com, accessed on 20 November 2021). Median MICs were bolded.

**Keratitis—*Staphylococcus aureus***								
**Antibiotic**	**Number**	**Minimum**	**Quarter1**	**Median**	**Quarter3**	**Maximum**	**MIC Susceptible**	**MIC Resistant**
Vancomycin	48	0.38	0.75	**0.75**	1.0	1.5	≤2	≥16
Gentamicin	48	0.094	0.19	**0.19**	0.24	4.0	≤4	≥16
Moxifloxacin	45	0.032	0.064	**0.094**	1.5	12.0	≤0.5	≥2
Cefoxitin	46	3.0	4.0	**4.0**	24.0	48.0	≤4	≥8
Cefazolin	47	0.38	0.75	**1.0**	3.0	96.0	No standard	No standard
**Keratitis—*Pseudomonas aeruginosa***								
**Antibiotic**	**Number**	**Minimum**	**Quarter1**	**Median**	**Quarter3**	**Maximum**	**MIC Susceptible**	**MIC Resistant**
Tobramycin	29	0.38	1.0	**1.5**	2.0	6.0	≤4	≥16
Ceftazidime	29	1.0	1.5	**1.5**	2.0	4.0	≤8	≥32
Ciprofloxacin	29	0.047	0.064	**0.125**	0.19	1.0	≤0.5	≥2
Polymyxin B	29	0.75	1.5	**2.0**	3.0	6.0	≤2	8
**Keratitis—*Coagulase Negative Staphylococci***								
**Antibiotic**	**Number**	**Minimum**	**Quarter1**	**Median**	**Quarter3**	**Maximum**	**MIC Susceptible**	**MIC Resistant**
Vancomycin	17	0.75	1.0	**1.5**	1.5	3.0	≤4	≥32
Gentamicin	17	0.03	0.08	**0.25**	19.0	48	≤4	≥16
Moxifloxacin	17	0.06	0.88	**3.0**	64.0	64.0	≤0.5	≥2
Cefazolin	17	0.38	0.5	**1.5**	3.0	12.0	No standard	No standard
**Keratitis—*Streptococcus viridans group***								
**Antibiotic**	**Number**	**Minimum**	**Quarter1**	**Median**	**Quarter3**	**Maximum**	**MIC Susceptible**	**MIC Resistant**
Vancomycin	13	0.25	0.44	**0.5**	0.88	2.00	≤1	No standard
Gentamicin	7	0.38	0.75	**2.0**	3.0	32.0	≤4	≥16
Moxifloxacin	13	0.064	0.094	**0.125**	0.19	0.038	≤1	≥4
Cefazolin	8	0.02	0.14	**0.32**	3.38	16	No standard	No standard
**Keratitis—Other Gram-positive bacteria**								
**Antibiotic**	**Number**	**Minimum**	**Quarter1**	**Median**	**Quarter3**	**Maximum**	**MIC Susceptible**	**MIC Resistant**
Vancomycin	8	0.38	0.41	**0.625**	1.0	1.5	≤1	No standard
Gentamicin	6	0.001	0.8	**2.5**	60	96.0	≤4	≥ 16
Moxifloxacin	8	0.014	0.211	**0.365**	1.28	4.0	≤1	≥ 4
cefazolin	8	0.001	0.1	**8.1**	4.2	128	No standard	No standard
**Keratitis—Other Gram-negative Bacteria**								
**Antibiotic**	**Number**	**Minimum**	**Quarter1**	**Median**	**Quarter3**	**Maximum**	**MIC Susceptible**	**MIC Resistant**
Tobramycin	14	0.25	0.348	**1.0**	3.0	3.0	≤4	≥16
Ceftazidime	13	0.064	0.125	**0.19**	1.0	3.0	≤8	≥32
Ciprofloxacin	13	0.012	0.019	**0.032**	0.47	4.00	≤1	≥4
Polymyxin B	12	0.001	0.56	**1.0**	6.38	32.00	≤2	≥8
**Conjunctivitis—*Staphylococcus aureus***								
**Antibiotic**	**Number**	**Minimum**	**Quarter1**	**Median**	**Quarter3**	**Maximum**	**MIC Susceptible**	**MIC Resistant**
Moxifloxacin	39	0.03	0.05	**0.06**	2.0	64.00	≤0.5	≥2
Cefoxitin	40	0.8	4.0	**4.0**	15.0	512	≤4	≥8
**Conjunctivitis—Other Gram-positive bacteria**								
**Antibiotic**	**Number**	**Minimum**	**Quarter1**	**Median**	**Quarter3**	**Maximum**	**MIC Susceptible**	**MIC Resistant**
Moxifloxacin	16	0.094	0.19	**0.25**	0.35	0.5	≤1	≥4
**Conjunctivitis—Other Gram-negative bacteria**								
**Antibiotic**	**Number**	**Minimum**	**Quarter1**	**Median**	**Quarter3**	**Maximum**	**MIC Susceptible**	**MIC Resistant**
Tobramycin	19	0.75	0.75	**1.5**	2.0	32	≤4	≥16
Ciprofloxacin	19	0.008	0.06	**0.047**	0.38	0.63	≤1	≥4
Polymyxin B	19	0.001	0.75	**1.0**	2.0	64	≤2	≥8
Moxifloxacin	13	0.047	0.094	**0.19**	0.94	6.0	No standard	No standard
**Endophthalmitis—Gram-positive bacteria**								
**Antibiotic**	**Number**	**Minimum**	**Quarter1**	**Median**	**Quarter3**	**Maximum**	**MIC Susceptible**	**MIC Resistant**
Vancomycin	25	0.5	1.0	**1.5**	2.0	2.0	≤4	≥32

## Data Availability

All of the data is included within the manuscript.

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
