# Peer review of "Anti-Infective Treatment and Resistance Is Rarely Problematic with Eye Infections"

_antibiotics, 2022, doi:10.3390/antibiotics11020204_

Round 1
Reviewer 1 Report
I have read the review with great interest. This is a well-written, detailed, and thorough review of topical anti-infective treatment and its resistance in cases of eye infections. I have no further suggestions or comments on it.
Author Response
I would like to thank the reviewer for their positive comment.
Reviewer 2 Report
The manuscript entitled “Anti-Infective Treatment and Resistance is Rarely Problematic with Eye Infections” is structured and written very well. The review presents provides the major insights in the use of antibiotics and resistance related issues for the treatment of eye diseases. However, it needs some major edits before acceptable for publication.
Line #35, other fluoroquinolone antibiotics also prescribed for bacterial keratitis.
https://pubmed.ncbi.nlm.nih.gov/17822972/.
doi: 10.1007/s40123-016-0046-6.
Line#31, Antibiotics and anti-infectives will be referred to as anti-infectives. Which category of anti-infectives refer in this statement. Specify?
Rewrite the statement with all fluoroquinolones used for the BK treatment.
In the early stages of diagnosis, ophthalmologists generally called as microbial infections either for fungal or bacterial infections. Write a brief note on the early stage diagnosis problems for the eye infections.
Line#46, include the subconjunctival injections also one of the treatment options with anti-infectives.
Eg. https://pubmed.ncbi.nlm.nih.gov/16141864/.
Cite the supportive references for line # 46-59.
Write a note on type of the disease (BE and BK) very highly effective in the population globally. Write a note on drugs used for BE and BK in the eye infections section.
Line 141, MIC – please specify is it MIC90 or MIC50 ?
The review lacks the significant references for the discussed statements in the eye diagnosis and treatment sections.
Line #313 – cite the reference for moxifloxacin (Moxeza®) formulations. Other novel formulations also reported for MOX.
https://pubmed.ncbi.nlm.nih.gov/33961956/.
https://www.ncbi.nlm.nih.gov/pmc/articles/PMC6571706/.
The nano delivery (SLNs, NLCs, NEs), in-situ gels, and other novel approaches (inserts, implants) also developed for the delivery of antibiotics to the back of the eye for treatment of eye infections. Discuss these approaches in the Alternative Anti-Infective Delivery and Therapy section.
https://www.ncbi.nlm.nih.gov/pmc/articles/PMC5874823/.
https://www.ncbi.nlm.nih.gov/pmc/articles/PMC7998883/.
Reviewer 3 Report
A sound and well written review.
Author Response
I would like to thank the reviewer for their positive comment
Reviewer 4 Report
Title
Title is specific and reflects the content of the manuscript; however, attention to rare ophthalmic antibiotic resistance and possible complications in the management of ocular infection should be mentioned to add more value to the manuscript. What happens with the antibiotic therapy before and after intravitreal injections? A specific mention to this very hot topic should be added. What happens when antibiotics are used for several times and short periods? What is the “relatioship” between the opinions expressed in the article and the ARMOR data?
Manuscript
Pag 2 line 46 – 59: Are there authors’ personal opinions? Please justify these sentences referring to peer-reviewed literature.
Pag 2 line 81 – 82 “The cornea, sclera, and conjunctiva have no colonizing bacteria.”: this sentence needs to be explained with references. Moreover, the presence of an ocular microbiome and microbiota is still uncertain and need to be proved. However, an increasing number of evidence suggest its presence.
Pag 3 line 86 – 87: The sentence “Any presence of pathogenic bacteria cultured from the cornea should be suspected as the etiologic agent” is misleading. Coinfections and superinfections need to be appropriately managed.
Pag 3 line 97 – 98: do you have any reference for this?
Pag 4 line 119 – 120: do you have any reference for this?
Pag 4 line 136: In this section authors analyze diagnostic of conjunctival infections; they report above the impact of adenoviral infections in conjunctivitis; a mention about point-of-care test for the diagnosis of adenoviral conjunctivitis may be appropriate. What about validity of these tests? NAAT? Please specify this acronym
Pag 5 line 180 – 185: empiric treatment for bacterial keratitis varies according to the clinical features of the ulcer. An empiric used of fortified antibiotics is usually reserved for large, high-risk, vision treating ulcers or for corneal infections not responding to conventional antibiotic therapy. Is there any role for fluoroquinolone monotherapy?
Pag 8 line 241 – 243: do you have any reference for this?
Pag 8 line 248-249: even if it is a rare event, some bacterial infections aren’t self-limiting and some of them may require hospitalization and systemic treatment for preventing serious complications (e.g., H. influenzae or N. gonorrhoeae).
References
Reference list includes several appropriate authors’ self-citations. Overall, they are relevant and significative according to the purpose of this review. Adding some relevant missed references cited by in literature about the main aspects presented in the manuscript is appropriate. Authors need to indicate the methods how they conduct literature research in this review.
Round 2
Reviewer 2 Report
Thank you for providing the responses for the clarification.
Author Response
It appears that we answered the comments of the second reviewer ....We Thank the reviewer for their positive comments.....
Reviewer 4 Report
Dear authors, it is not my intention to verify your expertise in this field.
I believe that aim of the manuscript could be also to give indications to ophthalmologists that are not fully involved in microbiology field and so, in my opinion, every strong (even if appropriate!) statement should be addressed to published literature.
Moreover, in this “hands-on methodology manuscript” the relationship with consistent literature reporting about the antibiotic resistance of ocular microorganisms (ARMOR and TRUST reports) should be addressed.
